# Coral-Focused Climate Change Adaptation and Restoration Based on Accelerating Natural Processes: Launching the "Reefs of Hope" Paradigm

**Austin Bowden-Kerby**

Corals for Conservation, P.O. Box 4649 Samabula, Suva, Fiji; abowdenkerby@gmail.com

**Abstract:** The widespread demise of coral reefs due to climate change is now a certainty, and investing in restoration without facing this stark reality risks failure. The 50 Reefs Initiative, the dominant adaptation model for coral reefs is examined, and a new coral-focused paradigm is proposed, based on helping coral reefs adapt to rising temperature, to ensure that as many coral species as possible survive locally over time. With pilot sites established in six Pacific Island nations, genebank nurseries of bleaching resistant corals are secured in cooler waters, to help prevent their demise as heat stress increases. Unbleached corals selected during bleaching events are included. From these nurseries corals are harvested to create nucleation patches of genetically diverse pre-adapted corals, which become reproductively, ecologically and biologically viable at reef scale, spreading out over time. This "Reefs of Hope" paradigm, modelled on tropical forest restoration, creates dense coral patches, using larger transplants or multiple small fragments elevated on structures, forming fish habitat immediately. The fish help increase coral and substratum health, which presumably will enhance natural larval-based recovery processes. We also hypothesize that incoming coral recruits, attracted to the patch, are inoculated by heat adapted algal symbionts, facilitating adaptation of the wider reef. With global emissions out of control, the most we can hope for is to buy precious time for coral reefs by saving coral species and coral diversity that will not likely survive unassisted.

**Keywords:** coral reef restoration; coral reef adaptation; coral bleaching resistance; assisted natural recovery; assisted gene flow; nature-based solutions; Reefs of Hope; 50 Reefs Initiative; Corals for Conservation

## 1. Introduction: Preventing Coral Reef Collapse

The collapse of coral reef ecosystems due to climate change over the next several decades is now considered a certainty. Predictions based on present trends indicate that by 2050, bleaching will become an annual occurrence on most reefs [1] and by that time only 10% of the world's reefs will persist [2] This collapse involves the extinction or near extinction of many coral species, even for corals found within reef systems presently considered refugia [3]. From a geologically functional perspective, recent findings on reef calcification rates indicate that few of the world's coral reefs will maintain their accretionary roles without the near-term stabilization of atmospheric $CO_2$ emissions [4].

Even under best-case emissions trajectories, coral reefs will continue to be negatively transformed by climate change [5,6]. Conventional coral reef restoration is unsuitable under climate change because increasing temperature stress must now be accepted as an established parameter of the environment, which will continue to impact newly outplanted corals during restoration [7].

Without the introduction of meaningful adaptive variation, there is a mismatch in the speed of adaptation relative to climate change [8], leading to local extirpation and limiting the long-term persistence of reefs [9]. Returning a degraded coral reef to its pristine state was once a realistic goal, but in many locations, it now appears that priorities must shift to supporting ecosystems that are more resilient to climate change, even if they represent

modified versions of the ideal state. Fortunately, the existing coral restoration toolbox is diverse and potentially adaptable to proactive restoration objectives [7,10].

Coral reefs in Kiribati in the central Pacific, and the Chagos Archipelago in the Indian Ocean, are on the leading edge of the predicted global collapse of coral reefs and have already experienced the biological or ecological extinction of multiple *Acropora* coral species due to marine heat waves lasting for months on end [3,11,12].

The time window for preventing the tipping point for the extinction of coral species is rapidly closing, with 10–30% of reefs predicted to remain if temperature rise is limited to 1.5 °C, but with only 1% remaining if the increase reaches 2 °C by the end of the century [13]. With the present lack of reduction in carbon release, in spite of the many promises, the planet is now on course to 3.2 °C, according to the 2022 IPCC report. "By 2050, the window of opportunity to act, both on coral reef adaptation and on climate change mitigation, will have closed for good." [14]. "While a massive effort, a "moonshot" approach, is appropriate for certain technological aspects of the solutions required, much of what needs to be done is more akin to the challenge of controlling a global pandemic, which depends on actions taken at thousands of locations around the world in a coordinated effort." [14].

In spite of promising signs of resiliency and recovery on reefs with few local stressors, mass bleaching threatens even the best managed and most pristine coral reefs. There is increasing evidence that reducing local stressors alone will not be sufficient to avert the rapid decline of coral reefs, due to the overwhelming impacts of increasing marine heat waves [15]. Even the best managed reefs have faced major coral declines in the face of mass bleaching, and MPAs have had no general effect on preventing coral loss or accelerating post-bleaching recovery [16].

If predictive scenarios developed for 1.5 °C, are readjusted to 3.2 °C, the situation begins to look hopelessly bleak. Investing considerable time and funds to carry out major coral restoration programs without facing the stark reality of out-of-control warming and increasing ocean temperatures risks losing opportunities for more successful adaptive approaches. To continue to plant corals for restoring reefs without recognizing what is coming is like continuing to plant trees for reforestation in the face of an approaching firestorm. A new approach to restoration in the face of what otherwise amounts to mass incineration is required, refocusing efforts on preventing the extinction of vulnerable coral species for at least a few more decades, while unrestrained climate change is hopefully brought under control.

Coral reef restoration efforts have advanced in recent decades, and heat resistance is being incorporated into some of these efforts. However, thus far no formal coral-focused strategy has been proposed, designed to help coral reefs adapt to rapidly warming seas. Corals that are adapted to hot waters found in places such as the Red Sea or on shallow nearshore reefs have been found to have no problem acclimating to moderately cool waters; however, these same heat-adapted corals have a hard time adapting when the water becomes hotter than normal [17]. The most bleaching-resistant corals will become increasingly vulnerable to death as conditions of the future surpass tolerance limits for corals within such hot pocket reefs, even if these reefs presently contain the most heat-tolerant individuals [18]. Indeed, such reefs may already be in the range of 35–38 °C during summer, near the maximum thermal limit possible for any corals to survive [17], and so moving the corals into cooler waters is suggested as an adaptation option [17,18]. Indeed, restoration efforts will ultimately fail unless they source bleaching-resistant corals as brood stock [19], focusing proactive interventions on outplanting heat-tolerant corals to cooler locations, assisting the migration of corals from warmer to cooler areas [18]. Based on this information, as well as future projections of increasingly severe heat waves, it is critically important to intervene to prevent heat-adapted corals from dying out.

## 2. Reexamining Existing Climate Change Adaptation Models for Coral Reefs

The only coral reef adaptation strategy widely recognized thus far is the "50 Reefs Initiative" [20]. This initiative uses modelling to choose the coral reefs deemed most likely

to survive into the future, in order to prioritize international efforts on saving coral reef systems from collapse, in effect abandoning the more impacted coral reef systems. The 50 Reefs Initiative is now accepted at the highest levels of international decision making and governance in the UN, forming a basis for site selection for the Global Fund for Coral Reefs (GFCR) program. However, several new studies cast doubt on the model, questioning the selection process for refugia, as well as the future functionality of the refugia themselves [3,18,21]. A study just out [18], concludes that a narrow focus on protecting refugia is a flawed strategy, as it fails to protect the more stress-adapted corals that are the best sources of adaptive capacity for future conditions. Another 2022 study has cast doubt as to the existence of thermal refugia altogether, based on present rates of warming [3]. Might lessons learned at the present forefront of the collapse [12] help to prevent the collapse of other coral reefs?

The 50 Reefs Initiative, on examination, does not create any new strategies for conserving coral reefs, but rather it continues to rely on no-take areas and pollution control measures, even though it has been established that pristine waters and unfished reefs are no more resilient to mass coral bleaching than more impacted reef systems [15–18].

As the 50 Reefs model is influencing high-level decision making regarding projects and funding, if there are major shortcomings in the model, it is vital to identify them. While the 50 reefs approach has good merit, if we are going to use the model for decision making, the model should be well scrutinized for omissions or misinterpretations, so that it can be reconfigured by the original authors. The most important questions to ask are if the best parameters were chosen for the model, what assumptions were made about those parameters, and were these assumptions correct?

The 50 Reefs Initiative study [20] uses only five parameters, three of which are related to coral bleaching: bleaching history, recent bleaching, and projected bleaching. Less than a third of the model focuses on other factors, namely connectivity and cyclones. Ocean acidification projections are surprisingly not included in the model, neither are present reef conditions, or reefs with areas of proven heat resilience.

Specific suggestions and considerations for a reworking of the 50 Reefs Initiative model are given below:

(1) Ocean Acidification should be included as a parameter, as not all coral reef areas are predicted to fare equally [21,22].
(2) Present reef condition, plus levels of protection from overfishing and land-based threats should be included in the model, especially as the establishment of no-take areas and implementation of pollution control measures are presently the dominant strategies for building resilience to climate change on coral reefs. This represents a major contradiction between the dominant management and adaptation paradigms.
(3) Coral species diversity might play a role in the selection matrix, as the present model treats all coral reefs as being equal.
(4) Presence and extent of pre-adapted, bleaching-resistant, coral populations living in warm waters within shallow lagoons and reef flats and up-current from other major coral reef areas must be included [8], with the intention of translocating corals which are bleaching-resistant from the hotter reefs to cooler waters before marine heat waves become lethal to these heat adapted coral populations.
(5) Cyclones as a parameter must be reassessed. The 50 Reefs Initiative assumption is that cyclones are 100% negative, however, a positive overall impact has been demonstrated by several researchers [23–25]. Indeed, we have seen NOAA predictions of mass coral bleaching in our Fiji sites dissipate on multiple occasions due to cyclones passing nearby. While cyclones may indeed wipe out or damage tens or even hundreds of km of reef, at the same time they cool off a much wider swath of water. The Chagos Archipelago [11] and the reefs of the three far-flung island groups of Kiribati, are among the most bleaching-impacted of all coral reefs, and all are located on or very near the equator. These island groups experience absolutely no cyclones, due to

opposing Coriolis forces, preventing this cyclone-mediated release of excessive heat build-up [12].

(6) While bleaching history and future bleaching projections should certainly be included in the model, making the model 2/3 based on bleaching, while excluding other vital parameters, does not seem prudent. The present model includes "recent thermal conditions", defined as summer temperatures over the past two years, as a separate parameter. This appears to be double counting, as bleaching history is already included as a parameter in the model.

(7) Connectivity certainly is a valid parameter to include in the model, however, connectivity among coral genera varies greatly. Connectivity is driven by currents and is dependent on the presence of up-current larval sources available for recolonization [20]. About 80% of broadcast spawning genera, such as *Acropora* have short-lived larvae [26], and it is essential for them to take in zooxanthella within five to seven days after settlement because a failure to do so may lead to death. In contrast, brooding coral genera such as *Pocillopora* carry photosynthetic algae with them and can produce food along the way, so they are much longer-lived larvae. Resilience, meaning the ability of the reefs to recover their original population and species mix after multiple mass bleaching events, should reflect estimates of connectivity, and thus resilience might make a better predictor than theoretical models of connectivity [20]. Coral reefs without up-current sources of heat-adapted larvae are more vulnerable, and in theory, can be expected to undergo phase shifts away from broadcasting species and *Acropora* dominance and towards brooding species dominance over time, which is what we have seen in the field.

The Global Coral Reef Monitoring Network GCRMN, which is the primary monitoring system to assess the overall condition of coral reefs on the planet, uses coral cover as their standard, with no record of genera, and thus the phase shift from *Acropora* dominance has gone unnoticed. *Acropora* gives way to *Pocillopora* dominance [27,28] to *Porites* dominance [12,29] or *Montipora* dominance [30] on the more impacted coral reefs. In our Fiji sites, we see evidence for a cascading series of shifts from *Acropora* to *Pocillopora* as the intermediate state, and then finally to *Porites*. Rather than running away from heavily bleaching impacted coral reefs and considering them hopeless, these coral reefs can provide key lessons important to preventing these phase shifts from occurring. The active coral focused adaptation methods described below can help retain and restore *Acropora* corals on these reefs.

## 3. Developing a Coral-Focused Adaptation Model Using Bleaching Resistant Corals

A review of efforts at assisted evolution in plant and animal communities and various actions that can be directed towards assisted evolution for coral reef adaptation was presented in 2015 by [31]. Matz et al. (2020) [7] demonstrated that adaptive potential, defined as the ability of a coral reef to maintain high coral cover over an extended time period in spite of recurring stress, can be predicted based on two metrics: the reef's present-day temperature, and more importantly, the proportion of coral recruits coming in from warmer locations. The study suggested that assisted gene flow could be encouraged via translocation as an intervention to boost the adaptive potential of coral populations. Caruso et al. [8] made an excellent argument in 2021 that, despite the numerous unknowns and potential trade-offs, a focus on thermal tolerance in restoration efforts is critical because corals that do not survive bleaching cannot contribute to future reefs at all, and that restoration without regard to thermal tolerance is essentially a dead end as the world warms.

Corals for Conservation, our Fiji-based NGO, is actively implementing a coral focused adaptation program involving the local translocation of corals between habitats, from hot to cool, within coral species natural ranges. Our program is based on lessons learned from the mass die off of even the most heat-adapted coral populations on Kiritimati Atoll in 2015–16 [12]. Local hot to cool water translocation has become one of the main strategies in our "Reefs of Hope" sites in Fiji, with partnership sites located in seven Pacific Island

nations [32]. Our operational assumption, is that as marine heat waves increase in intensity, populations of corals and associated endosymbionts presently located at the upper limit of their heat tolerance range will perish. As the ocean warms, we predict that these most heat-adapted corals are jeopardized due to their present location within nearshore and restricted flow hot pockets, and are no more secure in the face of approaching heat waves than are corals located on the cooler parts of the wider reef system, as the increased heat is superimposed on existing thermal regimes. For a coral population already experiencing 34–36 °C in midsummer, even a one- or two-degree increase during a heat wave event might prove fatal.

The future of coral reefs and the wider adaptation process depends on these most bleaching resistant corals, and so the goal of the Reefs of Hope project is to secure as many species and genotypes as possible. As the source populations of the heat-adapted corals are located mostly inshore and thus in more contaminated, polluted, silty, and pathogen rich waters, not only are the corals heat-resistant, but they may also be more disease-resistant. Corals are secured by collecting and moving samples of corals found within hot pockets out to cooler waters locally, initially placing them into reef-based gene bank nurseries to prevent their demise.

This Reefs of Hope adaptation paradigm is designed to strengthen and complement existing strategies, not to replace them, helping amplify natural adaptation processes within conservation areas. We seek to raise the alarm within the wider coral reef conservation community that we are facing a race against time to secure pre-adapted bleaching resistant corals, before temperatures become lethal and mass mortality events occur, as has happened on reefs in the forefront of the collapse [12,33].

Managed relocation of corals has been considered by others, based on concerns about coral reef persistence under future climate change scenarios [34]. Translocation is considered a potential way to increase adaptive processes through "assisted gene flow" from adapted coral populations to less adapted coral populations [34,35].

The Caruso et al. (2021) [7] study is perhaps the most advanced in its proposed approach and detailed strategies for seeking out local heat-resistant corals for propagation and use in restoration, and as a means to prevent failure in restoration efforts as the oceans warm. The study calls its approach "proactive restoration", and while the authors go into detail on selection methods for heat resistance and discuss the strong evidence for the persistence of heat resistance subsequent to translocation, they fail to explore the deeper impacts of translocating heat-adapted corals on assisted gene flow and symbiont sharing for climate change adaptation.

Our Reefs of Hope model uses elements of coral restoration, but is not based on numbers of corals planted or area covered, but rather on securing diverse genotypes of pre-adapted bleaching-resistant corals through translocation into cooler water gene bank nurseries in areas of good water flow but protected from cyclone waves and currents by local reef features. Nursery-reared pre-adapted corals are trimmed to produce second generation corals for planting into dense "nucleation patches", a concept borrowed from forest restoration, where natural reproduction and gene flow processes are restored, and where natural resilience and recovery processes are amplified within the patches.

## 4. Support for Translocating Bleaching Resistant Corals

Researchers have found high variability in heat tolerance thresholds for corals over reef-wide scales, which gives hope for considerable adaptation potential to increasing temperatures [36]. Heat-adapted coral populations have been found within shallow restricted-flow environments [37] and heat tolerance is finally being considered as vitally important in the field of coral restoration, and as a factor in the selection of brood stock [7]. However, local-scale translocation from the standpoint of preventing the potential extinction of populations of pre-adapted corals does not seem to be a goal of any of the ongoing or theoretical projects.

Translocation of heat and stress-adapted corals has recently been analyzed by a team of scientists at high level, as an option for facilitating adaptive process in corals, with risks and benefits considered [38]. Managed relocation was categorized into three types: 1. assisted gene flow, which involves local translocation within only tens of kilometers; 2. assisted migration, with translocation hundreds of kilometers distant but within a species natural range; and 3. species introduction, which refers to relocation outside of the present natural range of a coral species. Local translocation for assisted gene flow is considered low risk and is being carried out actively in restoration efforts, although generally not as a systematic adaptation strategy. Translocation between distant coral reefs and at regional scales is considered riskier from the standpoint of biosecurity. Translocation from the standpoint of preventing the death of pre-adapted coral populations does not seem to be a goal of any ongoing projects, nor was it mentioned in the NAS analysis.

We have taken local translocation to a higher level in our sites, with heat adaptation (bleaching resistance) being the most important factor in the selection of coral nursery brood stock. For reefs with strong thermal gradients, corals are translocated from hot pocket nearshore reefs and incorporated within gene bank nurseries located in cooler water areas, followed in subsequent years by the translocation of the nursery-reared corals into even cooler reefs, where they exist. We see this as a primary adaptation strategy that might be incorporated into ongoing coral reef restoration and biodiversity conservation projects globally.

Based on the situation in Kiribati [12], saving hot pocket corals via translocation should be regarded as time sensitive, and the strategy needs to be considered for urgent implementation wherever such genetic resources still exist. In this context, translocation is being done primarily to save coral/algal genotypes from going extinct. Only after securing the genotype should the focus change to assisted gene flow [35,38], as a natural outcome of the translocation.

## 5. Seeking out Both Symbiont-Based and Host-Based Bleaching Resistance

Bleaching-resistant corals obtain their resilience via two pathways; symbiont and host-based resilience [19,39–41]. We hypothesize that *Acropora* and other non-brooding coral species located within hot pocket reefs, as they acquire their algal symbionts after settlement, would get their resilience primarily from the algal symbionts they acquire from the existing population of bleaching resistant corals, while for cooler reef areas, incoming coral juveniles will have a very low probability of acquiring resistant symbionts. Therefore, when bleaching hits cooler water reefs, the few unbleached corals remaining on these cooler reefs might be expected to have a higher likelihood of exhibiting host-based resilience than would unbleached corals of the hotter reef areas. Using this hypothesis, in order to acquire more coral genotypes with host-based resilience within our nurseries, in addition to collecting from hot pocket reefs, we also focus efforts on sampling the few unbleached survivors of mass bleaching on cooler reefs. For reefs with a known history of heat waves and severe bleaching mortality, we also focus on sampling unusually large corals which have survived for many years, because size is a proxy for age and thus resilience over decades of stress in corals.

Heat-adapted corals have been shown to retain their heat resilience over time in cooler nursey and out-planting settings [17,19,42,43]. Bleaching resilience has been maintained in nursery environments for multiple species, and thus can be used to construct bleaching-resistant coral nurseries for climate-adapted restoration [19]. Both host genotype and symbiont genotype is not only feasible in stock selection for reef restoration and adaptation, but it is the way forward [19], and thus has been incorporated within the Reefs of Hope strategies.

## 6. The Need for Biosecurity Measures for Local Coral Translocation, and Suggested Methods

The possibility for coral disease to move within environments and to move between sites needs to be minimized [44] and we have taken this into the Reefs of Hope program.

This issue has recently received a lot of attention due to Stony Coral Tissue Loss Disease in the Caribbean [44]. Where the disease is afflicting a particular reef area, recommendations are circulating widely online that all dive and snorkeling gear, equipment, wetsuits, and clothes be sterilized when moving between infected and non–infected sites [45].

When conducting translocation between reefs in a local area, the following quarantine precautionary measures are recommended: (1) Avoid translocating corals from reefs with obvious disease outbreaks between reefs; (2) Avoid bringing newly collected corals directly into major gene bank nurseries as a disease prevention measure.

After collection of corals, a preliminary nursery should be established, located near the original hot pocket coral collection area, to serve as a quarantine and testing site. Corals are grown within the quarantine site for a year, to see that they survive the summer unbleached and disease-free, and if they are thriving and not impacted, they are then "graduated"- moved to the gene bank nursery further out in cooler waters.

Moving corals a few km, within a local reef system, from nearshore reefs to offshore reefs, is generally not very risky, as the waters already mix during a typical tidal cycle. However, moving corals between reef systems, even within the same country, becomes riskier. However, longer-distance translocation might be required if a coral species has become extinct or very rare, translocated over greater distances in order to restore sexual reproduction. In perhaps the first example of between-reef translocation of corals, multiple genotypes of four species of *Acropora*, which had become locally extinct on Kiritimati Atoll in 2015–2016, were collected in 2018 on Tabuaeran Atoll, some 180 Km distant, and flown to Kiritimati, where they continue to thrive [12].

International translocation, with the associated permits and costs, might at some point be needed for heavily impacted nations, where coral species have gone locally extinct. However, moving corals from an adjacent nation where they still exist, to a nation where they have become locally extinct, should only be considered for reefs within the same bioregion, where the former coral fauna was essentially the same species composition as that of the donor reefs. Tuvalu corals to Kiribati, for example, might become a situation where this approach is justified, while more distant reefs, such as those of Fiji, have some important species differences. However, before considering cross border translocation, more extensive surveys searching for coral survivors on multiple islands locally might avoid the need for international reintroductions.

## 7. Post-Bleaching Predation Eliminates Resistant Coral Survivors, Preventing Adaptation

Individual variation in heat tolerance among corals underpins the potential for coral adaptation [36]. This is related to both the host and symbiont, based on the genetic inheritability of the host as well as symbiont acquisition. During extreme mass bleaching events, some corals die, while others remain unbleached or partially bleached, providing variation on which natural selection for bleaching resistance can occur over time. However, researchers have not yet seen evidence for the evolution of beaching resistance in areas of the Great Barrier Reef experiencing multiple years of bleaching stress and recovery [46]. After the mass bleaching and die-off of some 90% of corals in southern Fiji in 2000 [47], personal observations indicate that the mortality of the surviving bleaching resistant corals was very high due to predation. The mass dying off of corals had apparently caused the predator-to-prey ratio to become skewed, with a super-abundance of crown of thorns starfish (COTS) and other predators in relation to the diminished coral population, which may be a primary factor behind the lack of adaptation on coral reefs facing sequential bleaching stress. Based on this information, subsequent to mass bleaching events, a rather urgent need exists for intervention to rescue bleaching-resistant corals for inclusion within protected nurseries. Predator control measures might also be implemented, to help the unbleached heat-adapted corals survive so that they can contribute their genetics and symbionts to the future.

### 8. Focus on the Most Vulnerable of Coral Species to Prevent Ecosystem Collapse

The relative resilience of coral species to climate change varies between genera [48]. While some may have concluded that to maintain coral cover on reefs into the future, the focus should be on the most resilient species, such as *Porites* corals. However, *Porites*-dominated reefs are very different from the standpoint of geological, biological and ecological processes, as well as from the fisheries production standpoint. If the focus is to maintain coral biodiversity and to prevent local species extinctions over time due to warming seas, it is vital to focus on retaining the coral species most at risk of extinction, not those which are presently faring well or increasing in abundance in spite of multiple stressors. *Acropora* species, although they are the dominant corals of intact coral reef ecosystems of both the Indo-Pacific and Caribbean, are often the first group to become locally extinct or rare, resulting in a phase shift to other species [12,27,49]. Safeguarding these fastest growing and vital habitat-structuring species and preventing their local extinction is essential if coral reefs are to retain a semblance of their original structure and functionality into the future.

As long as intact up-current coral reef systems remain intact, *Acropora* is often the first group to recover via larval recruitment [50] however when *Acropora* larval sources become limited, a phase shift occurs, often dominated by brooding genera such as *Pocillopora* [27,28,49], which may under continued stress give way to *Porites* [29], or *Montipora* [30]. One focus, therefore, is to maintain intact, functionally reproductive populations of *Acropora* corals, so that down-current coral reef areas will continue to receive coral larvae. Consideration for the dominant local currents during the spawning season should therefore be made with respect to the strategic location of restored spawning populations of corals. As a decision making tool, an excellent satellite-based and computer-generated model of real-time day-to-day current estimates can be found on the Windy Sea Temperature page (https://www.windy.com/-Sea-temperature-sst).

### 9. Summary of Coral-Focused Adaptation Measures: Reefs of Hope Strategies

(1) Translocate bleaching-resistant corals of targeted declining species (*Acropora*), from hot nearshore reefs approaching the maximum thermal threshold for coral survival, rescued from where their continued survival is questionable, and moved to cooler reef areas where their survival in warming seas is more secure.

(2) Sample unbleached corals of targeted species during or shortly after severe bleaching events, to secure them from predators, and as a selection strategy for bleaching resistant brood stock.

(3) Establish gene bank nurseries where heat-adapted corals of multiple declining species are grown and maintained over multiple years in a secure environment, and where size is kept in check through harvesting corals for use in coral reef restoration for climate change adaptation.

(4) Harvest nursery-reared corals to create genetically diverse patches of bleaching-resistant corals on cooler outer reefs, to facilitate restoration of sexual reproduction process and the formation of coral larvae, encouraging crosses between known bleaching-resistant strains and with cool-adapted coral strains of the outer reefs.

(5) Establish patches of bleaching resistant corals among populations of bleaching-sensitive corals to encourage potential sharing of resistant algal symbionts with surrounding corals, thereby spreading resilience more widely throughout the coral reef systems, and potentially as a natural outcome of moderate bleaching events.

(6) For reefs of low coral cover due to past bleaching or cyclone events, establish patches of bleaching-resistant corals in order to create a strong settlement signal for incoming coral larvae [51,52] while serving as a source for inoculation of juvenile corals with bleaching-resistant algal strains, as many coral species acquire their algae only after settlement. The offspring of many corals (approximately 80% of broadcasting species) acquire symbiotic zooxanthellae from the natural environment [26]. It is essential for them to settle and take in zooxanthellae within five to seven days after formation, because a failure to do so may lead to death.

## 10. Modeling Coral Restoration Strategies on Tropical Forest Restoration, to Assist Natural Adaptation and Recovery Processes

Coral reefs are often called the "rainforests of the oceans" because they are among the richest marine ecosystems in species, productivity, biomass, structural complexity, and beauty. Similar to rain forests, coral reefs are based on intricate interactions between organisms. Both systems rely on structural frameworks built by a single group of living organisms: trees in forests and corals on reefs. Similar to rain forests, coral reefs also thrive in nutrient-poor habitats enabled via symbiosis between multiple species, and reliant on complex food chains which recycle essential nutrients efficiently, making reefs especially sensitive to any process that disrupts recycling [53].

These natural vulnerabilities of coral ecosystems pose challenges to the development of a coral-focused adaptation paradigm. Because trees are the most important structural component of forests and corals are the most important structural component of coral reefs, replanting corals to degraded reefs has been likened to reforestation of the ocean [54]. "A robust restoration framework for the coral reef environment should equally set its priorities as the enhancement of reef biodiversity and the maintenance of ecosystems critical functions, thus providing the platform on which sustainable reef restoration should progress." [54].

Coral-focused strategies for coral reef adaptation can build on lessons learned from restoring forest ecosystems, to develop creative methods for harnessing key ecological processes, such as predation, herbivory, and nutrient cycling, that facilitate nature-based recovery [55,56]. It is these same natural processes of restoration focused on facilitated recovery which the Reefs of Hope model uses to facilitate increased thermal tolerance within coral populations.

In ecologically-based forest restoration efforts, tropical forest ecosystems best recover by focusing on replanting larger tree seedlings into small patches, which in turn become recovery nuclei; attracting important community members such as birds, lizards, and rodents, which in turn facilitate succession [57]. The focus is less on area covered and number of trees planted, and more on restoring patches of functional habitat, creating a dynamic situation whereby functionality and diversity begins from day one. Starting with larger saplings to increase structural complexity and create habitat to attract birds and other species which recycle nutrients, remove insect predators, and bring in more diversity through seed dispersal [58].

For ecologically-based coral reef restoration, we have developed a similar model which uses larger coral transplants and relies on the "grazing halo" phenomenon [59] the process by which herbivorous fish or sea urchins shelter in corals, and extend their grazing efforts outward, creating a barren algal-free zone around the coral often several meters wide. This cleared zone, if located on hard substratum, would become coralline algae dominated settlement surfaces, ideal for the recruitment of coral larval [60]. These clean grazed areas would also likely be lower in coral pathogens, as macro algae trap silt and bacteria, and are known to serve as reservoirs for coral disease [61].

Key natural processes are thus incorporated into restoration design in order to exploit dynamic ecological forces that promote the recovery of coral reef ecosystems; controlling factors such as the size, density, diversity, and identity of transplanted corals, as well as site selection, and transplant design to restore positive feedback processes, or to disrupt negative feedback processes, in order to improve restoration success [56].

Unlike reforestation by commercial timber interests, which tend to plant large areas with small seedlings of few species, the field of coral restoration for climate change adaptation should be aligned with tropical forest restoration efforts, which seek to restore the entire diverse ecosystem to its natural functional state through accelerating natural processes, while facilitating the sharing of heat-resistant symbionts with incoming coral larvae through careful selection of bleaching-resistant brood stock. This approach requires restoring, as early as possible, the structural complexity essential as the habitat of key grazing species [62] as well as the complex associations of organisms essential for optimal coral

reef functioning; restoring nutrient cycling, predator control, control of algal overgrowth, and the restoration of effective reproductive and recruitment processes. It is via enhancing recruitment and reproductive processes, coupled with translocation of thermal tolerance, that adaptation can best be facilitated.

Multiple positive interactions occur on coral reefs between corals and other organisms, from herbivorous fish and sea urchins keeping competing overgrowth of seaweeds down, to crabs and amphipods and fish, preventing the predation of corals [63]. Key positive species interactions that may facilitate the restoration of corals can include trophic facilitation, mutualisms, long-distance facilitation, positive density-dependence, positive legacy effects, and synergisms between biodiversity and ecosystem function. However, few studies exist which quantify any of these various effects [63] Although little is known about the specifics of the multiple and complex relationships that structure healthy coral reefs, an assumption can be made that the sooner heat-adapted coral transplants can begin to function as intact coral reef habitat, attracting and harboring diverse fish, crustacean and other species, the sooner these multiple synergisms will begin to operate to facilitate both recovery and adaptation. Unfortunately, this realization appears to be missing from most coral-focused restoration projects, which impose man-made rules and technologies onto a complex ecosystem, rather than working with nature to reboot natural processes. While some may assert that more studies are needed before we can begin to engage in active coral-focused adaptation, the situation has become urgent, and so even if some of the processes and underlying assumptions are poorly understood, we may not need to understand all of the complexities and synergistic interactions in order to begin harnessing them. Trial and error, and learning by doing, has never been more appropriate.

## 11. Restoring Nutrient Cycling by Restoring the Fish–The "Birds of the Reef"

Bleaching and coral death causes drastic changes in fish trophic groups, particularly a decline in coral-dependent fish such as plankton-feeding species confined to living branching corals [64]. Abundant clouds of plankton-feeding fish can often be seen over living branching corals on many reefs, capturing nutrients from the water column and channeling them into the reef ecosystem. These fish soon disappear after their living branching coral habitats die. In nutrient-poor waters of the Indo Pacific, *Chromis* and *Dasyllus* fish are particularly important planktonivorous fish species which capture nutrients and pass them down the food chain to the wider coral reef ecosystem; into the corals through their wastes, and directly into the fish community as the fish are eaten by fish such as trumpet fish, groupers, and jacks. Mass coral death due to coral bleaching, and the impact of such a drastic event on planktonivorous fish has been recorded in several studies [64,65]. In the Caribbean, juvenile grunts, snappers, and other species, although they are not live coral obligates, often form dense schools over staghorn *Acropora*, serving a similar ecological function of adding nutrients to the corals, improving coral growth [66,67].

Just as the goal of tropical forest restoration is to utilize the synergistic effects of birds [58] a goal of ecological coral reef restoration should be to restore beneficial fish populations as soon as possible, to take advantage of the synergistic effects of the fish. Observations in our restoration sites indicate that using larger branching coral transplants taken from nurseries after two years of growth can provide immediate fish habitat [67], and can thereby shorten the time to greater ecological functionality of restoration patches by up to 2–3 years. Although no in-depth study yet exists, the restored fish and invertebrates within such patches of larger restored corals should result in better coral nutrition through trophic exchange, as described above, but also faster coral growth [67], disease resistance, and reproductive success. Juvenile and adult bottom feeding and algae eating fish would also create a foraging and grazing halo around the transplanted corals and improve substrate conditions for recruitment, which could in turn facilitate larval settlement. Lobsters moving into the transplants might benefit restored corals by feeding on snail predators which often attack outplanted corals [68]. Once *Stegastes* farmer damselfish move into patches of large staghorn corals, despite the obvious negative impact of basal area

death, as they convert live tissue into their garden territories, major causes of coral death would be reduced, such as COTS predation, snail predation, butterflyfish and parrotfish predation [12], and the smothering by large cyanobacteria drift algae such as *Lyngbya* [69].

## 12. Summary of Coral Restoration for Assisted Natural Recovery of Coral Reefs

Through the launching of the Reefs of Hope adaptation paradigm, coral reef restoration has been rechanneled to focus on securing vulnerable species of corals while facilitating natural recovery and adaptation processes. Working with nature, and inspired by new paradigms of tropical forest restoration, the strategy facilitates the establishment of multiple synergies between corals and the species that they provide a vital habitat for, resulting in greater biodiversity and vitality that is capable of spreading. An alternative to the prevailing passive approach to climate change adaptation for coral reefs is now available, with active coral-focused strategies, designed to re-open bottlenecks in coral recovery; assisting with post-bleaching coral survival, restoring reproduction and larval formation processes, enhancing coral recruitment to degraded reef areas, and encouraging the sharing of bleaching resistant symbionts. These active strategies can help ensure that as many coral species and genotypes as possible survive locally, while creating synergies between fish, corals, and other species, to help reboot natural recovery and adaptation processes. Global-scale replication of this intensive rather than extensive approach is possible on a patch by patch, reef by reef, and coral species by coral species basis. Just as importantly, with proper training in the methods and with minimal supervision and follow-up, these methods are understandable and thus appropriate for implementation at community level, as proven in multiple Pacific Island sites in Fiji and in six Pacific Island nations [32].

## 13. Conclusions

Until we solve the bigger threat of climate change by reducing carbon emissions, healthy, balanced, and diverse coral reefs will, one by one, be lost to warming seas. The most we can hope for is to buy precious time by slowing this demise. This can be achieved by ensuring that patches of diverse and climate-adapted corals of each species on coral reefs remain ecologically, reproductively, and biologically viable at reef scale. Rather than imposing restoration solutions on the reef, this strategy, modelled on tropical forest restoration ecology, seeks to restore patches of more complete coral communities, replete with protective fish and invertebrates, which in turn increase coral and substratum health and enhance natural recruitment, recovery, and adaptation processes on the wider reef. The underlying assumption to all of our efforts must be that climate change will eventually be brought under control.

The essential need to engage governments and the wider community of small island developing states, which are on the forefront of ecosystem collapse, in the solutions, even at this early and often experimental stage, cannot be overstated. Upscaling of the active coral focused measures described in this paper can occur through widespread involvement by governments, NGOs, the tourism industry, and reef-dependent communities.

**Funding:** UNEP, Plantation Island Resort (Fiji), Global Giving, Conservation Food and Health Foundation (USA), Just World Partnerships (UK), Southern Cross Cable, DFAT (via WWF and Kyemma Foundation), Ridge to Reef Tuvalu, FAO Samoa, and US Embassy Suva.

**Acknowledgments:** Field and scientific support was provided by Igor Pessoa, Sarah Makutu, Merekeleni Tinai, Annelise McDougall, Wilson Hazelman, Kirsty Firth, Jerome Mafutuna, Kolora Lewadradra, Cedric Rocholl, Laurence Romeo, Taratau Kirata, and Tavita Faletoese.

**Conflicts of Interest:** The author declares no conflict of interest.

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
