# Peer review of "Coral-Focused Climate Change Adaptation and Restoration Based on Accelerating Natural Processes: Launching the “Reefs of Hope” Paradigm"

_2673-1924, doi:10.3390/oceans4010002_

Round 1

Reviewer 1 Report

A central theme of this paper is that corals pre-adapted to heat stress should be moved to cooler water to prevent extirpation. However, the paper should acknowledge the possibility that such coral will lose heat resistance. Morikawa & Palumbi (2019) says that coral won’t lose this resistance, but in fact we do not yet know.

This paper does a good job of summarizing a lot of recent literature. One recent paper that may be relevant is Van Woesik et al 2022 (see below).

Introduction is repetitive. Paragraph on 54 – 63 could be deleted, or key elements (first sentence and reforestation analogy) moved elsewhere.

Paragraph 200 -216 is repetitive and can be reduced or eliminated. See also 226 – 230.

313 For transplantation purposes, it is important to define ecoregions. Perhaps use the classification by Kulbicki et al (2013). I think the point about international boundaries is almost irrelevant (reminiscent of ‘geopolitical species’; Copus et al. 2018). It’s the ecoregions that matter, not the political barriers.

Paragraph 396 – 403 is beautifully written but not necessary. This whole section on outplanting (394 – 441) seems like a side trip, and may be reduced or eliminated. Key points can be incorporated into the next section. Most already are.

References are a mess. Some of this might be a simple format issue, but I found citations to missing references starting in the second paragraph:

 Drury et al., 2017a; Drury and Lirman, 2021.

Minor Points:

Acropora, Pocillopora, Montipora, and Porites are genera, and should be italicized.

89 What is GFCR program?

153 Delete ‘While’.

162 This statement could use a citation.

191 Delete ‘As’

284 Delete ‘on’.

330 Define ‘COTS’.

346 Not a complete sentence.

435 – 438 Sentence is awkward. Please reword.

481 Possibly use ‘harnessing’ instead of ‘taking advantage of’. Same on 499.

485 – 487 This sentence can be removed.

518 Lyngbya is also a genus and should be italicized.

560 ‘author’, singular

Typos: lines 50, 85, 91, 96, 313, 342, 490 mostly punctuation.

Overall this is a valuable commentary. I don’t agree with all the points, but the author has a right to express them because they are certainly well-informed and thought-provoking. The primary problem is the repetition. This paper can be cut by a third without loss of information, and I have tried to point out some places where this can be accomplished.

Possibly Useful References:

Copus et al 2018. Geopolitical species revisited: Genomic and morphological data indicate that the Roundtail Chub Gila robusta species complex (Teleostei, Cyprinidae) is a single species. PeerJ 6:e5605

Kulbicki et al (2013) Global biogeography of reef fishes: a hierarchical quantitative delineation of regions. PLoS ONE, 8, e81847.

Timmers et al 2021. Biodiversity of coral reef cryptobiota shuffles but does not decline under the combined stressors of ocean warming and acidification. Proceedings of the National Academy of Sciences 118(39):e2103275118.

Van Woesik et al 2022. Coral‐bleaching responses to climate change across biological scales. Global Change Biology 28(14):4229-50.

Author Response

I have reworked sections based on your advice, and it has greatly improved the paper, so thanks very much for doing such a super job at reviewing the paper.  You helped the most of anyone!  I have done most of what you suggested, and more. The Forest stuff is now much better integrated with the paper too, and I was able to cut some sections way down and get rid of repetition.  

I added some key references, and am still going over the citations/references, to make sure they are all there and in order.   

Austin

Author Response

Please see the attachment. I found additional references and added them.  I also reworked several sections.

Thanks very much!   

Austin

Reviewer 3 Report

This commentary on the future of coral reefs addresses shortcomings of the parameters chosen for the “50 Reefs Model” and assumptions made on these parameters. The authors propose that the model include sections on:

a.     OA

b.     Present reef condition

c.     Coral diversity

d.     Parameters of corals (pre-adapted, bleaching resistant, etc.)

e.     Effect of cyclones

f.      Factoring in things besides bleaching history and future projections

g.     Connectivity (= resilience)

These comments could be at the beginning of useful comments on improving the predictions on the “50 Reefs Model”.

Then they go on to suggest development of an “adaptation model using bleaching resistant corals”, which basically involves moving corals out of “hot pockets” to cooler water. 

Questions: What are you waiting for? Governments to enact lower emissions? The local temperatures to decrease to more tolerable levels? The corals to adapt to higher temperatures?

It is proposed that translocation of corals can be over “tens of kilometers”, “hundreds of kilometers”, or “outside of the present natural range”. These transfers are to cooler reefs, “if they exist”. It seems that the potential is to exist now, but what do the projections look like for the future? Not good.

It seems to me that for any of this to work, there must be symbiont algal types that can function at the warm temperatures. There have been a couple types that can survive in hot water (but only a few). It might be good to concentrate efforts on finding heat-resistant symbionts that can survive in a particular coral host.

The other problem with heat-stressed corals is that they seem to get infected with all sorts of nasty bacteria. This is probably not their fault; in heating up seawater humans have made the conditions better for bacteria. I’m not sure anything can be done about that?

Variation in the response to heated seawater is the basis for coral adaptation. But if the temperature keeps getting warmer there is a limit to coral/symbiont resilience. There should be a concerted effort to determine the characteristics of resistant corals (e.g. biomass, symbiont densities, symbionts types) vs heat-sensitive corals. There is a lot to be learned by monitoring and knowing exactly what is happening.

Treating coral reefs like rain forests is easier said the done. If you follow restoration of reefs in South Florida you will know that it is much more difficult (and expensive) to transplant corals compared to trees/plants in forests. 

I’ve never heard of anyone restoring fish to a reef, but I guess it is something to think about. There are lots of problems that need to be overcome, not the least of which is how to deal with the top-predator (= humans).

I am all in for this, and any, approach to conserve coral reefs. But as the author says, “the underlying assumption to all of our efforts must be that climate change will eventually be brought under control”. The question is when? Before or after coral reefs succumb. 

Author Response

Please see the attached.  I have re-worked the paper based on input of the other reviewers, I also added a bit on disease resistance based on your comment. 

Sorry it looks so messy with the track changes! 

Thanks,

Austin 

Round 2

Reviewer 3 Report

Changes are fine.